# Comorbidity Profile of Chronic Mast Cell–Mediated Angioedema Versus Chronic Spontaneous Urticaria

**DOI:** 10.3390/biomedicines13092259

**Published:** 2025-09-13

**Authors:** Eli Magen, Iris Leibovich, Israel Magen, Eugene Merzon, Ilan Green, Avivit Golan-Cohen, Shlomo Vinker, Ariel Israel

**Affiliations:** 1Leumit Health Services, Tel Aviv-Yafo 6473817, Israel; emarzon@leumit.co.il (E.M.); igreen@leumit.co.il (I.G.); agolanchoen@leumit.co.il (A.G.-C.); svinker@leumit.co.il (S.V.); aisrael@leumit.co.il (A.I.); 2Medicine A Department, Assuta Ashdod University Medical Center, Ben Gurion University of the Negev, Beer Sheva 8410501, Israel; 3Faculty of Health Sciences, Ben-Gurion University of the Negev, Beer Sheva 8410501, Israel; irisl@bmc.gov.il (I.L.); isr.magen@gmail.com (I.M.); 4Allergy, Clinical Immunology, and Angioedema Unit, Barzilai University Medical Center, Ashkelon 7830604, Israel; 5Adelson School of Medicine, Ariel University, Ariel 4070000, Israel; 6Department of Family Medicine, Gray Faculty of Medical and Health Sciences, Tel-Aviv University, Tel Aviv-Yafo 6997801, Israel; 7Department of Epidemiology and Preventive Medicine, School of Public Health, Faculty of Medical & Health Sciences, Tel Aviv University, Ramat Aviv, Tel Aviv-Yafo 6139001, Israel

**Keywords:** mast cell–mediated angioedema, chronic spontaneous urticaria, comorbidity, cardiovascular risk, retrospective cohort, statins, chronic urticaria phenotype

## Abstract

**Background:** Chronic mast cell–mediated angioedema (MC-AE) and chronic spontaneous urticaria (CSU) both involve mast cell activation but may differ in long-term systemic outcomes. Limited data exist comparing their comorbidity profiles over extended follow-up. **Objective:** To compare systemic comorbidities in patients with chronic MC-AE versus CSU using a large, population-based dataset. **Methods:** We conducted a retrospective matched case–control study using electronic health records from Leumit Health Services, a nationwide Israeli health maintenance organization. Patients diagnosed with chronic MC-AE between 2005 and 2023 (*n* = 2133) were matched 1:1 by age, sex, and year of diagnosis to patients with CSU (*n* = 2133). Comorbidities were assessed at diagnosis and after a mean follow-up of 10.2 ± 2.9 years. Odds ratios (ORs) with 95% confidence intervals (CIs) were calculated. Multivariable logistic regression was used to assess the association between medications and MC-AE diagnosis. **Results:** MC-AE patients exhibited significantly higher baseline rates of hypertension (23.8% vs. 18.5%), ischemic heart disease (5.67% vs. 3.84%), and type 2 diabetes (10.45% vs. 6.42%) compared to CSU. These differences persisted or increased at follow-up, including myocardial infarction (4.13% vs. 2.25%) and chronic kidney disease (4.13% vs. 2.91%). CSU patients had consistently higher rates of atopic dermatitis, viral infections, and herpes zoster. Statin use was inversely associated with MC-AE (adjusted OR = 0.63; 95% CI: 0.44–0.90). **Conclusions:** Chronic MC-AE is associated with a distinct and sustained cardiometabolic and renal comorbidity burden compared to CSU, supporting its classification as a systemic disease phenotype requiring differentiated long-term care.

## 1. Introduction

Chronic mast cell–mediated angioedema (MC-AE) is a distinct clinical entity characterized by recurrent, localized episodes of subcutaneous or submucosal edema, driven primarily by mast cell activation and histamine release [1]. Unlike bradykinin-mediated forms, which are typically unresponsive to antihistamines and often associated with hereditary or acquired C1 inhibitor deficiency, MC-AE arises from immunologically mediated mast cell degranulation [2]. It is frequently responsive to H1 antihistamines or anti-IgE therapy [2,3]. Despite its relative prevalence in clinical practice, chronic MC-AE remains under-investigated as an independent phenotype, particularly in terms of its systemic comorbidities and disease burden.

Chronic spontaneous urticaria (CSU) and chronic MC-AE share central pathogenic mechanisms involving dysregulated mast cell activation and mediator release [4]. However, clinical and epidemiological observations suggest that angioedema-predominant phenotypes may represent a pathophysiologically and prognostically distinct subgroup [5]. Nonetheless, there remains a paucity of high-quality, population-based studies characterizing the broader clinical profile of patients with chronic histaminergic angioedema compared to those with CSU.

In a previous study, we investigated the clinical and laboratory features that distinguish MC-AE from antihistamine-responsive CSU, revealing that low IgE in MC-AE and higher IgE in CSU may signify two distinct types of immune dysregulation [6].

Mounting evidence suggests a role for mast cell–derived mediators, including histamine, tryptase, platelet-activating factor, and various cytokines, in promoting vascular inflammation, endothelial dysfunction, insulin resistance, and immune dysregulation [7]. These mechanisms may underlie increased rates of cardiometabolic, renal, autoimmune, and infectious comorbidities observed in mast cell–associated diseases [8]. However, whether patients with chronic MC-AE exhibit a unique comorbidity pattern distinct from CSU remains unknown.

To address this gap, we conducted a population-based case–control study comparing patients with chronic MC-AE to age- and sex-matched controls with CSU. Utilizing comprehensive clinical data from a nationwide health maintenance organization in Israel, we aimed to delineate the differential prevalence of cardiometabolic, renal, endocrine, allergic, and infectious comorbidities.

## 2. Materials and Methods

### 2.1. Study Design and Data Source

We conducted a retrospective, population-based, matched case–control study utilizing anonymized electronic health records (EHRs) from Leumit Health Services (LHS), an Israeli health maintenance organization that insures approximately 750,000 individuals. The LHS centralized database integrates longitudinal medical, laboratory, and prescription data from both primary care and specialist visits, coded according to the International Classification of Diseases, Ninth Revision (ICD-9), and linked to pharmacy dispensation records using Anatomical Therapeutic Chemical (ATC) codes.

### 2.2. Study Population

Cases were defined as patients diagnosed with chronic MC-AE, identified between 2005 and 2023. Inclusion criteria included: (1) documentation of angioedema persisting for ≥6 weeks, (2) absence of concurrent urticarial wheals, (3) therapeutic response to H1-antihistamines or anti-IgE therapy, and (4) exclusion of bradykinin-mediated etiologies (e.g., hereditary or acquired C1-INH deficiency) through clinician documentation or diagnostic testing. Only patients with ICD-9 codes 995.1 (Angioedema), confirmed by manual chart review or specialist diagnosis, were included.

Control subjects were randomly selected from the patients diagnosed with CSU within LHS between 2005 and 2023. Inclusion criteria for CSU controls included a diagnosis of CSU (ICD-9: 708.x, excluding inducible urticarias) persisting for ≥6 weeks, with or without accompanying angioedema. Patients with isolated angioedema were excluded from the CSU cohort. To ensure representative sampling, potential controls were selected from the complete CSU registry and matched 1:1 with cases based on age (within ±1 year), sex, and year of diagnosis. Randomization was performed using a computerized random number generator in R software (version 4.3.1), ensuring each eligible CSU patient had an equal probability of selection. If a match could not be found within the specified age/sex criteria, the case was excluded to maintain strict matching integrity.

Comorbidities were assessed at two time points: (1) at the time of diagnosis (baseline) and (2) at the end of the observation period (31 December 2023). The mean follow-up duration from the date of MC-AE or CSU diagnosis to the end of the observation period was 10.2 ± 2.9 years.

### 2.3. Demographic and Clinical Variables

We extracted demographic data (age, sex, and socioeconomic status), anthropometric indices (weight, height, and BMI), and smoking status. Comorbidities were identified using established ICD-9 definitions across six major disease domains: cardiovascular, renal, metabolic/endocrine, allergic/dermatologic, infectious, and autoimmune conditions.

### 2.4. Medication Exposure

Pharmacological exposures were assessed using ATC-coded dispensation data within the 12 months preceding the index diagnosis. Medication classes of interest included statins (C10AA), antihypertensives (ACE inhibitors, ARBs, beta-blockers, and calcium channel blockers), corticosteroids (H02AB), and antidiabetics (metformin and insulin). Only chronic users (≥3 dispensations over ≥6 months) were classified as exposed.

### 2.5. Statistical Analysis

Descriptive statistics were computed for all variables. Categorical variables were compared using chi-square tests, and continuous variables using Student’s *t*-test or Mann–Whitney U test as appropriate. Standardized mean differences (SMDs) were calculated to assess group balance. Odds ratios (ORs) with 95% confidence intervals (CIs) were estimated for each comorbidity using univariable logistic regression.

Multivariable logistic regression was used to assess the association between medication use and the odds of being classified in the MC-AE versus CSU group, while adjusting for all concurrent comorbidities. False discovery rate (FDR) adjustment was applied for multiple comparisons using the Benjamini–Hochberg method. All statistical analyses were conducted using R software version 4.3.1, and a two-sided *p*-value < 0.05 was considered statistically significant.

### 2.6. Ethics Approval

The study protocol was approved by the Leumit Research Institute Ethics Committee (IRB number: LEU 05-23). Data were de-identified before analysis, and individual patient consent was waived due to the study’s retrospective nature and the use of anonymized records.

## 3. Results

### 3.1. Demographics

The study included 4266 patients, comprising 2133 individuals diagnosed with MC-AE and 2133 matched controls with CSU. The gender distribution was identical between the groups, with 59.0% females and 41.0% males in both cohorts (*p* = 1.000, OR = 1.00 [95% CI: 0.88–1.13]). The mean age of patients was similar between the groups (37.7 ± 22.6 years in MC-AE vs. 37.5 ± 22.6 years in CSU; *p* = 0.759).

Body mass index (BMI) was significantly higher in the MC-AE group (25.6 ± 6.6 vs. 24.9 ± 6.1; *p* = 0.001, SMD = 0.100). Obesity (BMI ≥ 30) was significantly more prevalent among MC-AE patients (23.1% vs. 19.0%, *p* = 0.001; OR = 1.28 [95% CI: 1.10–1.50]). The higher prevalence of obesity in MC-AE compared with CSU was largely driven by middle-aged patients (30–59 years).

No significant group differences were observed in smoking status and physical activity.

The socio-economic status (mean SES score of ~9.5 in both groups) was comparable between the groups (Table 1).

### 3.2. Comorbidities at Baseline

#### 3.2.1. Cardiovascular Diseases

Hypertension was significantly more prevalent in the MC-AE group (508; 23.8%) compared to the CSU group (395; 18.5%), *p* < 0.001, OR = 1.37 [1.18–1.60]. Ischemic heart disease (5.67% vs. 3.84%; *p* = 0.006, OR = 1.50 [1.12–2.03]) and myocardial infarction (2.48% vs. 1.08%; *p* < 0.001, OR = 2.34 [1.40–4.01]) were also more prevalent in patients with angioedema. Major adverse cardiovascular events (MACE) occurred in 4.17% of MC-AE vs. 2.58% of CSU patients (*p* = 0.005, OR = 1.64 [1.16–2.36]).

#### 3.2.2. Renal Diseases

The prevalence of chronic kidney disease (1.03% vs. 0.89%) and microalbuminuria (8.81% vs. 7.17%) was slightly higher in MC-AE, with the latter approaching statistical significance (*p* = 0.055, OR = 1.25 [1.00–1.57]).

#### 3.2.3. Endocrine and Metabolic Diseases

Type 2 diabetes mellitus was significantly more frequent among MC-AE patients (10.45% vs. 6.42%; *p* < 0.001, OR = 1.70 [1.36–2.14]). No significant differences were found for dyslipidemia or autoimmune thyroid disorders, though Graves’ disease showed a trend (1.50% vs. 0.89%, *p* = 0.090).

#### 3.2.4. Allergic and Dermatologic Disorders

Food allergy was diagnosed in 58 (2.72%) MC-AE and 79 (3.70%) CSU patients (OR 0.73, 95% CI 0.52–1.03, *p* = 0.082), while systemic anaphylaxis was noted in 21 (0.98%) and 29 (1.36%) patients, respectively (OR 0.72, 95% CI 0.41–1.27, *p* = 0.319.

MC-AE patients had a lower prevalence of atopic dermatitis (9.19% vs. 12.00%; *p* = 0.003), contact dermatitis (23.6% vs. 27.9%; *p* = 0.002), dermatophytosis (25.2% vs. 31.0%; *p* < 0.001), and pityriasis rosea (1.13% vs. 1.92%; *p* = 0.045).

#### 3.2.5. Infectious Diseases

Bacterial pneumonia was more common in MC-AE patients (3.09% vs. 1.97%; *p* = 0.025, OR = 1.59 [1.06–2.41]). In contrast, CSU patients had higher rates of tonsillitis (43.6% vs. 40.3%; *p* = 0.030), viral infections (54.2% vs. 49.9%; *p* = 0.005), viral warts (19.7% vs. 16.4%; *p* = 0.005), and herpes zoster (5.53% vs. 3.09%; *p* < 0.001) (Table 2).

### 3.3. Medication Use Against MC-AE and CSU

As shown in Table 3, systemic corticosteroid prescriptions were slightly less common in MC-AE than CSU (43.0% vs. 47.2%; OR 0.84, 95% CI 0.75–0.95; *p* = 0.006), while long-term corticosteroid use was rare and similar in both groups. Sustained H1-antihistamine use was modestly lower in MC-AE (93.1% vs. 95.5%; OR 0.64, 95% CI 0.49–0.83; *p* = 0.001). Leukotriene receptor antagonists and omalizumab were also used significantly less often in MC-AE.

### 3.4. Comorbidities After 10 Years of Follow-Up

#### 3.4.1. Cardiovascular Diseases

Hypertension remained more prevalent in MC-AE (32.9%) compared to CSU (28.1%; *p* < 0.001, OR = 1.25 [1.10–1.43]). Rates of ischemic heart disease (9.14% vs. 7.13%; *p* = 0.019), myocardial infarction (4.13% vs. 2.25%; *p* < 0.001), PCI (3.47% vs. 2.20%; *p* = 0.016), CHF (4.74% vs. 3.23%; *p* = 0.015), and MACE (6.66% vs. 4.50%; *p* = 0.003) were all significantly higher in MC-AE.

#### 3.4.2. Renal Diseases

Chronic kidney disease (4.13% vs. 2.91%; *p* = 0.037, OR = 1.44 [1.02–2.03]) and microalbuminuria (10.50% vs. 8.49%; *p* = 0.028, OR = 1.27 [1.02–1.56]) were significantly more prevalent in patients with MC-AE.

#### 3.4.3. Endocrine and Metabolic Diseases

T2DM was more prevalent in MC-AE (17.9% vs. 13.3%; *p* < 0.001, OR = 1.42 [1.20–1.69]), along with dyslipidemia complications (1.78% vs. 1.03%; *p* = 0.050). Thyroid diseases were comparable between groups.

#### 3.4.4. Allergic and Dermatologic Disorders

Food allergy was documented in 71 MC-AE patients (3.32%) and 93 CSU patients (4.36%) (OR 0.76, 95% CI 0.54–1.05, *p* = 0.094), while systemic anaphylaxis occurred in 27 (1.27%) versus 42 (1.96%) patients, respectively (OR 0.64, 95% CI 0.38–1.06, *p* = 0.089). Atopic dermatitis (13.2% vs. 17.9%; *p* < 0.001), contact dermatitis (34.9% vs. 42.5%; *p* < 0.001), dermatophytosis (41.3% vs. 48.4%; *p* < 0.001), and pityriasis rosea (1.97% vs. 3.38%; *p* = 0.006) remained significantly more common in CSU.

#### 3.4.5. Infectious Diseases

CSU patients continued to show higher prevalence of tonsillitis (57.7% vs. 54.1%; *p* = 0.021), viral infections (70.2% vs. 66.8%; *p* = 0.018), viral warts (33.5% vs. 29.0%; *p* = 0.002), and herpes zoster (10.74% vs. 8.25%; *p* = 0.007) (Table 4).

### 3.5. Medication Use and Adjusted Odds of MC-AE

In the multivariable logistic regression analysis adjusted for comorbidities (including cardiovascular, metabolic, allergic, dermatologic, infectious diseases, and autoimmune conditions), statin use was significantly associated with reduced odds of MC-AE (adjusted OR = 0.63, 95% CI: 0.45–0.90, *p* = 0.012). Conversely, insulin use was associated with a substantially elevated, but not statistically significant, odds of MC-AE (adjusted OR = 2.21; 95% CI, 0.76–6.46; *p* = 0.148). No statistically significant associations were found for systemic corticosteroids, ACE inhibitors, angiotensin receptor blockers, beta-blockers, calcium channel blockers, or metformin (Table 5).

### 3.6. Stepwise Model

In the stepwise-selected logistic regression model, the strongest independent predictors of MC-AE were hypertension (OR 1.24, 95% CI 1.09–1.42, *p* < 0.001), type 2 diabetes mellitus (OR 1.40, 95% CI 1.19–1.66, *p* < 0.001), myocardial infarction (OR 1.83, 95% CI 1.26–2.65, *p* = 0.002), chronic kidney disease (OR 1.41, 95% CI 1.01–1.99, *p* = 0.044), and obesity (BMI ≥30) (OR 1.28, 95% CI 1.12–1.47, *p* < 0.001). Negative associations were observed for atopic dermatitis (OR 0.71, 95% CI 0.60–0.84, *p* < 0.001), contact dermatitis (OR 0.74, 95% CI 0.65–0.83, *p* < 0.001), dermatophytosis (OR 0.77, 95% CI 0.67–0.87, *p* < 0.001), and herpes zoster (OR 0.76, 95% CI 0.61–0.94, *p* = 0.010). Vitiligo remained an independent positive predictor (OR 1.55, 95% CI 1.22–1.95, *p* < 0.001).

## 4. Discussion

This study highlights a persistent and divergent comorbidity trajectory in patients with chronic MC-AE versus CSU over a 10-year period. MC-AE was associated with cardiometabolic and renal conditions, while CSU was linked to atopic and infectious comorbidities.

At baseline, MC-AE was associated with higher rates of hypertension, ischemic heart disease, myocardial infarction, and type 2 diabetes mellitus, reflecting an early cardiometabolic vulnerability. These associations persisted and, in some cases, intensified over time, suggesting a potential chronic vascular or metabolic dysfunction. In contrast, CSU patients demonstrated a consistently higher prevalence of atopic dermatitis and certain infectious conditions. Nevertheless, our findings indicate that the higher infectious burden in CSU is attributable mainly to viral and fungal infections, while bacterial infections showed only a modest and transient baseline association. Thus, the association of CSU with infections should be interpreted cautiously, as it is not uniform across all pathogen types. These findings support previous hypotheses of divergent immunophenotypes between CSU and MC-AE, with CSU potentially representing a Th2-skewed, eosinophil-driven disease, and MC-AE representing a more localized mast cell activation with systemic sequelae [9,10]. The higher prevalence of cutaneous and mucosal infections in CSU may be mediated by skin barrier dysfunction, altered microbiome composition, or differential use of immunosuppressive treatments [11,12,13]. Chronic subclinical mast cell activation in MC-AE may represent a sustained inflammatory and fibrotic stimulus, predisposing individuals to macrovascular atherosclerotic events and myocardial dysfunction, and offering a plausible mechanistic explanation for the elevated rates of IHD and CHF observed in this cohort [14]. The emergence of renal dysfunction as a significant comorbidity in MC-AE patients is particularly notable. It is plausible that subclinical, mast cell–mediated endothelial damage in the renal microvasculature accumulates over time, eventually manifesting as chronic kidney disease [15,16]. Mast cells have been identified in glomerular and tubulointerstitial infiltrates in various renal pathologies, and their mediators, particularly histamine and TGF-β, have been implicated in promoting glomerulosclerosis and tubulointerstitial fibrosis [17,18,19].

Interestingly, asthma—classically considered a mast cell–driven disease—did not differ between the groups, suggesting that the systemic consequences of mast cell activation in MC-AE extend along a more vasculometabolic axis, whereas CSU’s comorbid profile remains more dermoatopic and mucosal. This dichotomy may also reflect differences in mast cell activation pathways: while CSU is predominantly driven by spontaneous activation via autoreactive IgG or IgE, MC-AE may involve alternative activation routes, such as MRGPRX2 stimulation or complement-derived anaphylatoxins (e.g., C3a, C5a) [20,21,22]. The absence of statistically significant associations for food allergy and systemic anaphylaxis, both at baseline and after 10 years, may be explained by several factors. First, the relatively low prevalence of these conditions in our cohort likely limited statistical power to detect modest differences. Second, whereas food allergy and anaphylaxis are mediated by allergen-specific IgE responses, MC-AE and CSU are thought to arise primarily from mast cell activation that occurs independently of allergen-specific IgE [20,21,22]. This fundamental divergence in immunopathogenesis may account for the lack of a strong association between MC-AE and IgE-mediated disorders. Larger prospective studies with detailed allergological phenotyping are needed to clarify this observation.

The significant association of statin use with reduced odds of MC-AE adds an intriguing translational layer. Statins possess anti-inflammatory, endothelial-stabilizing, and mast cell–modulating properties, including inhibition of FcεRI expression, reduction in mast cell degranulation, and suppression of pro-inflammatory cytokine release [23,24]. These effects may partially account for their protective role observed here, independent of their lipid-lowering capacity. By contrast, ACE inhibitors, although well known to induce bradykinin-mediated angioedema, showed no association with the chronic, mast cell–mediated phenotype examined in this cohort, supporting mechanistic heterogeneity within angioedema syndromes [25]. Collectively, our findings advance the understanding of MC-AE as a systemic immunoinflammatory disorder rather than a localized cutaneous phenomenon. The chronicity and breadth of comorbidity accumulation—spanning cardiovascular, metabolic, and renal domains—suggest a need for proactive longitudinal monitoring and integrative care pathways. In contrast, CSU may warrant enhanced surveillance for infectious and atopic complications, and perhaps stratification based on endotype-specific biomarkers such as D-dimer, IL-31, or autoreactive IgE [26].

The analysis of treatment patterns indicates that the higher prevalence of metabolic and cardiovascular comorbidities in MC-AE cannot be attributed to greater medication exposure. On the contrary, long-term corticosteroid use was infrequent and comparable, while leukotriene receptor antagonists and omalizumab were prescribed significantly less often in MC-AE than in CSU.

We observed the paradox that statin use was lower in MC-AE despite higher cardiometabolic comorbidity. Our data did not provide a clear explanation; we can only speculate that differences in prescribing priorities, or adherence may contribute. This highlights a potential treatment gap requiring further study.

These results argue strongly for reframing chronic MC-AE as a distinct clinical entity, with implications for both pathophysiological research and clinical management. Risk-based screening protocols for cardiovascular disease and metabolic dysfunction should be considered in this population, potentially guided by inflammatory and endothelial biomarkers (e.g., hs-CRP, sVCAM-1, IL-6) [27]. Direct biomarker studies in MC-AE are lacking; however, convergent evidence from CSU demonstrates increased levels of chemokines (CXCL8/IL-8, CXCL9/10, CCL2, CCL5) [28], (CCL11, CCL17, CCL26, CCL27) [29] and adhesion molecules (P-selectin, ICAM-1, VCAM-1) [30,31], which decrease with effective therapy, indicating mast-cell–driven endothelial activation [32]. These pathways are not limited to cutaneous inflammation: soluble adhesion molecules and chemokine axes are well-established markers of endothelial dysfunction, with prospective links to hypertension, type 2 diabetes, myocardial infarction, chronic kidney disease, and obesity [33]. It is therefore biologically plausible that recurrent mast-cell activation in MC-AE, through endothelial priming and chemokine-mediated leukocyte recruitment, contributes to the distinct cardiometabolic and renal comorbidity burden observed in our cohort [34].

At baseline, both psoriasis and vitiligo were infrequent and not significantly associated with either phenotype, likely reflecting the small numbers observed. After 10 years, psoriasis remained unassociated, maybe duo to its predominant Th17/IL-23–driven biology [35], whereas vitiligo, characterized by type I interferon/CD8^+^ T cell–mediated autoimmunity [36], emerged as a significant comorbidity in MC-AE. Although this finding currently lacks a definitive mechanistic explanation, it may reflect the previous observation that mast cells might be involved in vitiligo induction and progression [37]. Accordingly, this should be regarded as a hypothesis-generating observation that warrants confirmation in future prospective studies.

Although direct data on MC-AE are lacking, hereditary angioedema is increasingly linked to alterations in the gut microbiome [38]. Reviews further highlight gut dysbiosis as a common feature across chronic inflammatory skin diseases and support a causal link to CSU [39,40]. These findings support exploring the microbiome as a potential driver of the distinct systemic comorbidity burden in MC-AE.

This study’s strengths include its large, population-based design, strict diagnostic criteria, and long follow-up period, which enable a robust comparison of MC-AE and CSU over time. Matching on age, sex, and diagnosis year, along with manual chart review, enhanced diagnostic accuracy and minimized bias. However, limitations include potential misclassification due to reliance on ICD-9 codes, lack of systematic data on lifestyle factors, and inability to confirm medication adherence. Although physical activity was available and included in our analysis, other important lifestyle factors—including diet, alcohol intake, tobacco use, and family history of cardiometabolic disease—are not systematically captured in the LHS electronic records and therefore could not be incorporated. The potential influence of disease severity and control on comorbidity burden must also be considered, but in our cohort validated severity indices such as UAS7 or AE-QoL were unavailable. Nonetheless, future prospective studies with standardized severity and lifestyle assessments are needed to validate our findings. As a retrospective observational study, causality cannot be inferred from the observed associations, and prospective interventional studies are needed to confirm these observations.

Future mechanistic studies should focus on dissecting the transcriptomic and proteomic profiles of circulating and tissue-resident mast cells in MC-AE vs. CSU, and exploring how chronic activation signatures translate into organ-specific pathology. Furthermore, randomized controlled trials examining the potential of mast cell–targeted therapies (e.g., omalizumab, dupilumab, BTK inhibitors, or statins) to mitigate systemic comorbidity risks are warranted.

## Figures and Tables

**Table 1 biomedicines-13-02259-t001:** Demographic and Clinical Characteristics.

Variable		MC-AE (*n* = 2133)	CSU (*n* = 2133)	*p*-Value
Female, *n* (%)		1258 (59.0%)	1258 (59.0%)	1.000
Age (years), mean ± SD		37.7 ± 22.6	37.5 ± 22.6	0.759
AGE category (years)	0–2	81 (3.79%)	100 (4.69%)	0.171
	3–9	207 (9.71%)	195 (9.14%)	0.564
	10–18	265 (12.42%)	278 (13.03%)	0.581
	19–29	320 (15.01%)	299 (14.02%)	0.385
	30–39	239 (11.21%)	242 (11.35%)	0.923
	40–49	270 (12.66%)	271 (12.71%)	1.000
	50–59	302 (14.16%)	318 (14.91%)	0.515
	60–69	285 (13.36%)	263 (12.33%)	0.337
	70–79	127 (5.95%)	134 (6.28%)	0.702
	80–89	36 (1.69%)	33 (1.55%)	0.808
BMI (kg/m^2^), mean ± SD		25.6 ± 6.6	24.9 ± 6.1	0.001
BMI ≥ 30, *n* (%)		462 (23.1%)	386 (19.0%)	0.001
Past smoker, *n* (%)		41 (2.28%)	55 (3.03%)	0.179
Current smoker, *n* (%)		344 (19.1%)	309 (17.0%)	0.110
Socioeconomic status, mean ± SD		9.50 ± 3.56	9.54 ± 3.56	0.739
Creatinine (mg/dL)		0.75 ± 0.35	0.74 ± 0.22	0.185
eGFR (mL/min/1.73 m^2^)		126 ± 103	128 ± 106	0.597
Glucose (mg/dL)		96.2 ± 23.3	94.5 ± 20.6	0.014
HbA1c (%)		5.71 ± 0.92	5.60 ± 0.86	<0.001
HDL (mg/dL)		51.9 ± 14.4	52.2 ± 13.9	0.570
LDL (mg/dL)		108 ± 34	110 ± 34	0.027
Non-HDL cholesterol (mg/dL)		134 ± 39	136 ± 39	0.080
Hemoglobin (g/dL)		13.3 ± 1.5	13.4 ± 1.4	0.112
Physical activity	1–3 h weekly	473 (24.95%)	496 (26.22%)	0.372
	>3 h weekly	188 (9.92%)	168 (8.88%)	0.290
	None	631 (33.28%)	569 (30.07%)	0.036
	Occasionally	604 (31.86%)	659 (34.83%)	0.054
	Missing data	237 (11.11%)	241 (11.30%)	0.884

**BMI**—Body Mass Index, **eGFR**—Estimated glomerular filtration rate, **HDL**—High-density lipoprotein cholesterol, **LDL**—Low-density lipoprotein cholesterol, **non-HDL cholesterol**—Total cholesterol minus HDL.

**Table 2 biomedicines-13-02259-t002:** Comorbidities at Baseline.

Category	Comorbidity	MC-AE *n* (%)	CSU *n* (%)	*p*-Value	OR (95% CI)
**Cardiovascular Diseases**	Hypertension	508 (23.8%)	395 (18.5%)	<0.001	1.37 [1.18–1.60]
	Ischemic heart disease	121 (5.67%)	82 (3.84%)	0.006	1.50 [1.12–2.03]
	Myocardial infarction	53 (2.48%)	23 (1.08%)	<0.001	2.34 [1.40–4.01]
	PCI	42 (1.97%)	27 (1.27%)	0.089	1.57 [0.94–2.65]
	CABG	25 (1.17%)	15 (0.70%)	0.152	1.57 [0.94–2.65]
	CHF	22 (1.03%)	21 (0.98%)	0.999	1.05 [0.55–2.01]
	CVA	29 (1.36%)	27 (1.27%)	0.893	1.08 [0.61–1.89]
	MACE	89 (4.17%)	55 (2.58%)	0.005	1.64 [1.16–2.36]
**Renal Diseases**	Chronic kidney disease	22 (1.03%)	19 (0.89%)	0.754	1.05 [0.55–2.01]
	Microalbuminuria	188 (8.81%)	153 (7.17%)	0.055	1.25 [1.00–1.57]
**Endocrine &** **Metabolic** **Diseases**	T2DM	223 (10.45%)	137 (6.42%)	<0.001	1.70 [1.36–2.14]
	Dyslipidemia	535 (25.1%)	527 (24.7%)	0.804	1.02 [0.89 –1.18]
	Graves’ disease	32 (1.5%)	19 (0.89%)	0.090	1.69 [0.93–3.17]
	Hashimoto’s thyroiditis	94 (4.41%)	109 (5.11%)	0.314	0.86 [0.64–1.15]
**Allergic &** **Dermatologic** **Diseases**	Allergic rhinitis	274 (14.39%)	307 (12.85%)	0.153	0.90 [0.76–1.07]
	Asthma	157 (7.36%	159 (7.45%)	0.953	0.99 [0.78–1.25]
	Food allergy	58 (2.72%)	79 (3.70%)	0.082	0.73 [0.52–1.03]
	Systemic anaphylaxis	21 (0.98%)	29 (1.36%)	0.319	0.72 [0.41–1.27]
	Atopic dermatitis	196 (9.19%)	256 (12.0%)	0.003	0.74 [0.61–0.91]
	Contact dermatitis	503 (23.6%)	595 (27.9%)	0.002	0.80 [0.70–0.92]
	Psoriasis	56 (2.63%)	40 (1.88%)	0.121	1.41 [0.92–2.18]
	Vitiligo	7 (0.33%)	13 (0.61%)	0.262	0.54 [0.18–1.45]
	Dermatophytosis	538 (25.2%)	661 (31.0%)	<0.001	0.75 [0.65–0.86]
	Pityriasis rosea	24 (1.13%)	41 (1.92%)	0.045	0.58 [0.33–0.99]
**Infectious diseases**	Bacterial pneumonia	66 (3.09%)	42 (1.97%)	0.025	1.59 [1.06–2.41]
	Tonsillitis	860 (40.3%)	930 (43.6%)	0.030	0.87 [0.77–0.99]
	Viral infections (general)	1064 (49.9%)	1156 (54.2%)	0.005	0.84 [0.75–0.95]
	Cytomegalovirus	21 (0.99%)	18 (0.84%)	0.748	1.17 [0.62–2.20]
	Viral warts	350 (16.4%)	420 (19.7%)	0.005	0.80 [0.68–0.93]
	Herpes zoster	66 (3.09%)	118 (5.53%)	<0.001	0.55 [0.39–0.75]
**Connective tissue diseases**		44 (2.06%)	48 (2.25%)	0.915	0.91 [0.59–1.41]

**CABG**—Coronary artery bypass grafting, **CHF**—Congestive heart failure, **CVA**—Cerebrovascular accident, **MACE**—Major adverse cardiovascular events, **PCI**—Percutaneous coronary intervention.

**Table 3 biomedicines-13-02259-t003:** Prevalence of Long-Term Medication Use in Patients with MC-AE and CSU.

Medication Class	MC-AE (*n* = 2133)	CSU (*n* = 2133)	Odds Ratio (95% CI)	*p*
**Systemic corticosteroid (any prescription)**	917 (43.0%)	1007 (47.2%)	0.84 (0.75–0.95)	0.006
**Systemic corticosteroids (≥3 months cumulative use)**	102 (4.8%)	90 (4.2%)	1.14 (0.85–1.52)	0.417
**H1-antihistamines (any prescription)**	2133 (100%)	2133 (100%)	–	1.000
**H1-antihistamines (daily use ≥ 3 months)**	1986 (93.1%)	2037 (95.5%)	0.64 (0.49–0.83)	0.001
**H2-antihistamines (≥3 months)**	28 (1.3%)	31 (1.5%)	0.90 (0.54–1.51)	0.793
**Leukotriene receptor antagonists**	92 (4.3%)	172 (8.1%)	0.51 (0.40–0.67)	<0.001
**Omalizumab (≥6 months use)**	0 (0.0%)	42 (2.0%)	0.00 (0.00–0.19)	<0.001
**Other immunosuppressants (cyclosporine)**	0 (0.0%)	7 (0.3%)	0.00 (0.00–1.16)	0.016

**Table 4 biomedicines-13-02259-t004:** Comorbidities After 10 Years of Follow-Up.

Category	Comorbidity	MC-AE *n* (%)	CSU *n* (%)	*p*-Value	OR (95% CI)
**Cardiovascular** **Diseases**	Hypertension	702 (32.9%)	599 (28.1%)	<0.001	1.25 [1.10–1.43]
	Ischemic heart disease	195 (9.14%)	152 (7.13%)	0.019	1.31 [1.05–1.65]
	Myocardial infarction	88 (4.13%)	48 (2.25%)	<0.001	1.87 [1.29–2.73]
	PCI	74 (3.47%)	47 (2.20%)	0.016	1.59 [1.09–2.36]
	CABG	35 (1.64%)	28 (1.31%)	0.447	1.25 [0.74–2.15]
	CHF	101 (4.74%)	69 (3.23%)	0.015	1.49 [1.08–2.06]
	CVA	67 (3.14%)	61 (2.86%)	0.654	1.10 [0.76–1.59]
	MACE	142 (6.66%)	96 (4.5%)	0.003	1.51 [1.15–2.00]
**Renal Diseases**	Chronic kidney disease	88 (4.13%)	62 (2.91%)	0.037	1.44 [1.02–2.03]
	Microalbuminuria	224 (10.50%)	181 (8.49%)	0.028	1.27 [1.02 –1.56]
**Endocrine & ** **Metabolic Diseases**	T2DM	382 (17.9%)	284 (13.3%)	<0.001	1.42 [1.20–1.69]
	Dyslipidemia	38 (1.78%)	22 (1.03%)	0.050	1.74 [1.00–3.10]
	Graves’ disease	54 (2.53%)	50 (2.34%)	0.766	1.08 [0.72–1.63]
	Hashimoto’s thyroiditis	163 (7.64%)	194 (9.10%)	0.097	0.83 [0.66–1.03]
**Allergic &** **Dermatologic** **Diseases**	Allergic rhinitis	514 (24.1%)	538 (25.20%)	0.414	0.94 [0.82–1.08]
	Asthma	218 (10.2%)	220 (10.30%)	0.960	0.99 [0.81–1.21]
	Food allergy	71 (3.32%)	93 (4.36%)	0.094	0.76 [0.54–1.05]
	Systemic anaphylaxis	27 (1.27%)	42 (1.96%)	0.089	0.64 [0.38–1.06]
	Atopic dermatitis	282 (13.2%)	382 (17.9%)	<0.001	0.70 [0.59–0.83]
	Contact dermatitis	744 (34.9%)	907 (42.5%)	<0.001	0.73 [0.64–0.82]
	Psoriasis	99 (4.64%)	91 (4.27%)	0.603	1.09 [0.81–1.48]
	Vitiligo	103 (1.12%)	332 (0.72%)	<0.001	1.56 [1.23–1.95]
	Dermatophytosis	881 (41.3%)	1032 (48.4%)	<0.001	0.75 [0.66–0.85]
	Pityriasis rosea	42 (1.97%)	72 (3.38%)	0.006	0.58 [0.38–0.86]
**Infectious Diseases**	Bacterial pneumonia	98 (4.59%)	81 (3.8%)	0.222	1.22 [0.89–1.67]
	Tonsillitis	1154 (54.1%)	1231 (57.7%)	0.021	0.87 [0.77–0.98]
	Viral infections (general)	1425 (66.8%)	1497 (70.2%)	0.018	0.85 [0.75–0.97]
	Cytomegalovirus	39 (1.83%)	37 (1.73%)	0.912	1.06 [0.67–1.66]
	Viral warts	619 (29.0%)	715 (33.5%)	0.002	0.81 [0.71–0.93]
	Herpes zoster	176 (8.25%)	229 (10.74%)	0.007	0.75 [0.60–0.92]
**Connective tissue diseases**		90 (4.22%)	81 (3.80%)	0.532	1.12 [0.81–1.54]

**CABG**—Coronary artery bypass grafting, **CHF**—Congestive heart failure, **CVA**—Cerebrovascular accident, MACE—Major adverse cardiovascular events, **PCI**—Percutaneous coronary intervention.

**Table 5 biomedicines-13-02259-t005:** Medication Use and Adjusted Odds of MC-AE (Comorbidity-Adjusted Model).

Medication	Adjusted OR [95% CI]	*p*-Value
**ACE inhibitors**	0.97 [0.67–1.40]	0.862
**ARBs**	1.14 [0.77–1.68]	0.510
**Beta blockers**	1.00 [0.63–1.59]	0.990
**Calcium channel blockers**	1.60 [0.88–2.89]	0.124
**Statins**	0.63 [0.44–0.90]	0.012
**Metformin**	1.21 [0.79–1.86]	0.390
**Insulin**	2.21 [0.76–6.46]	0.148
**Systemic corticosteroids**	0.81 [0.49–1.36]	0.429

## Data Availability

The data supporting the findings of this study are available from Leumit Health Services, but restrictions apply to the availability of these data, which were used under license for the current study and are not publicly available. Data are, however, available from the corresponding author upon reasonable request and with permission of Leumit Health Services.

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
