# Peer review of "Comorbidity Profile of Chronic Mast Cell–Mediated Angioedema Versus Chronic Spontaneous Urticaria"

_biomedicines, 2025, doi:10.3390/biomedicines13092259_

Round 1

Reviewer 1 Report

Comments and Suggestions for Authors

The study, 'Comorbidity Profile of Chronic Mast Cell–Mediated Angioedema Versus Chronic Spontaneous Urticaria,' is a very interesting and well-written manuscript. Given that Chronic Mast Cell–Mediated Angioedema (CMA) and Chronic Spontaneous Urticaria (CSU) are closely related in their mast cell-mediated pathophysiology, with hives being the key clinical distinction, this large matched case-control study provides novel insights by highlighting differences in their comorbidity profiles.

I have a few queries and suggestions to further enhance the manuscript:

  1. Role of Medication in Comorbidities: The study reports a higher prevalence of long-term metabolic diseases, such as diabetes mellitus, hypertension, and cardiovascular disease, in the CMA group compared to the CSU group. It would be beneficial to investigate whether this difference could be a long-term effect of treatment, particularly corticosteroid use. I suggest that the analysis be expanded to include the use of specific medications, such as antihistamines and corticosteroids, to explore their potential influence on the comorbidity profiles.

  2. Discrepancy in Statin Use: The finding that the odds ratio for statin use is lower in the CMA group—despite a higher prevalence of cardiovascular disease and related risk factors like diabetes—requires clarification. Please discuss this apparent discrepancy and offer a potential explanation.

  3. Impact of Disease Characteristics on Comorbidities: The manuscript would be strengthened by an analysis that explores the relationship between disease characteristics and comorbidities. Specifically, were comorbidities influenced by the severity of the disease, the level of disease control, or the type and duration of medication use?

Author Response

The study, 'Comorbidity Profile of Chronic Mast Cell–Mediated Angioedema Versus Chronic Spontaneous Urticaria,' is a very interesting and well-written manuscript. Given that Chronic Mast Cell–Mediated Angioedema (CMA) and Chronic Spontaneous Urticaria (CSU) are closely related in their mast cell-mediated pathophysiology, with hives being the key clinical distinction, this large matched case-control study provides novel insights by highlighting differences in their comorbidity profiles.

I have a few queries and suggestions to further enhance the manuscript:

  1. Role of Medication in Comorbidities: The study reports a higher prevalence of long-term metabolic diseases, such as diabetes mellitus, hypertension, and cardiovascular disease, in the CMA group compared to the CSU group. It would be beneficial to investigate whether this difference could be a long-term effect of treatment, particularly corticosteroid use. I suggest that the analysis be expanded to include the use of specific medications, such as antihistamines and corticosteroids, to explore their potential influence on the comorbidity profiles.

Response:

We thank the reviewer for this important suggestion regarding the possible influence of long-term medication use on comorbidity profiles. As recommended, we performed a detailed analysis of prescription and dispensing data from the Leumit Health Services electronic records, focusing on systemic corticosteroids, antihistamines, leukotriene receptor antagonists, omalizumab, and immunosuppressants. The results are summarized in the new Table 3.

Our findings demonstrate:

  • Systemic corticosteroid use (any prescription) was actually less frequent in MC-AE than in CSU (43.0% vs. 47.2%; OR 0.84, 95% CI 0.75–0.95; p = 0.006). Importantly, long-term corticosteroid use (≥3 months cumulative exposure) was infrequent in both groups (<5%) and did not differ significantly (4.8% vs. 4.2%; OR 1.14, 95% CI 0.85–1.52; p = 0.417). Thus, corticosteroid exposure does not appear to account for the higher prevalence of metabolic and cardiovascular comorbidities observed in CMA.
  • H1-antihistamines were universally prescribed in both groups. However, sustained daily use for ≥3 months was slightly less frequent in MC-AE compared with CSU (93.1% vs. 95.5%; OR 0.64, 95% CI 0.49–0.83; p = 0.001). This may suggest differences in disease characteristics or treatment responsiveness but is unlikely to explain the comorbidity differences.
  • H2-antihistamines (1.3% vs. 1.5%; OR 0.90, 95% CI 0.54–1.51; p = 0.793) and long-term immunosuppressants (rare overall, ≤0.3%) showed no meaningful differences.
  • Leukotriene receptor antagonists (4.3% vs. 8.1%; OR 0.51, 95% CI 0.40–0.67; p < 0.001) and omalizumab (0.0% vs. 2.0%; OR 0.00, 95% CI 0.00–0.19; p < 0.001) were less commonly used in MC-AE compared with CSU.

Taken together, these analyses indicate that the higher burden of metabolic and cardiovascular disease in MC-AE cannot be explained by greater exposure to corticosteroids, antihistamines, or other anti-mast-cell therapies. On the contrary, corticosteroid and advanced biologic use were lower in MC-AE than in CSU. This strengthens the interpretation that MS-AE represents a distinct phenotype with an inherently higher systemic comorbidity burden, independent of pharmacological treatment.

  1. Discrepancy in Statin Use: The finding that the odds ratio for statin use is lower in the CMA group—despite a higher prevalence of cardiovascular disease and related risk factors like diabetes—requires clarification. Please discuss this apparent discrepancy and offer a potential explanation.

Response:

We thank the reviewer for this insightful observation. Indeed, we also noted the paradox that statin use was lower in CMA despite a higher prevalence of cardiometabolic comorbidities. After analyzing our dataset, we could not identify a definitive explanation for this discrepancy. Traditional determinants of statin initiation, such as LDL and non-HDL cholesterol, were not substantially different between groups, and adjustment for these variables did not resolve the paradox.

Therefore, we can only speculate on possible contributing factors. These may include differences in diagnostic coding or physician prescribing behaviors, or patient-level factors such as adherence. Importantly, this observation highlights a potential treatment gap in cardiovascular risk management for CMA patients that warrants further prospective study.

We have now addressed this limitation and added a corresponding statement to the Discussion.

  1. Impact of Disease Characteristics on Comorbidities: The manuscript would be strengthened by an analysis that explores the relationship between disease characteristics and comorbidities. Specifically, were comorbidities influenced by the severity of the disease, the level of disease control, or the type and duration of medication use?

Response:

We thank the reviewer for this valuable suggestion. Unfortunately, validated disease severity indices such as UAS7 (Urticaria Activity Score), AE-QoL (Angioedema Quality of Life), or physician global assessments are not systematically recorded in the Leumit Health Services EHR and therefore could not be incorporated into our analysis. Similarly, direct measures of disease control were unavailable. We have acknowledged this limitation in the manuscript with the following statement: “The potential influence of disease severity and control on comorbidity burden must be considered; however, in our cohort validated severity indices such as UAS7 or AE-QoL were unavailable. Future prospective studies with standardized severity and control measures are warranted to validate these findings.”

Reviewer 2 Report

Comments and Suggestions for Authors

While the study is well defined, further clarification is needed regarding methodological aspects. Being a retrospective analysis, there is a possibility of misclassification and missing confounding factors that were not measured. In addition, using more advanced statistical methods such as improved model selection, classification curve (ROC), reclassification of models using tools like Dtreg , or survival analysis could help provide a clearer picture of the key factors that are truly linked with MC-AE.

There are a few comments that require clarification from the authors, along with some suggestions that could help make this study more informative.

  1. The authors have described various baseline variables and their associations; however, important lifestyle factors (such as diet, physical activity, and alcohol use) as well as family history were not captured. These factors could confound the observed cardiometabolic outcomes.

  1. In Table 1 (Demographic and Clinical Characteristics), the difference in BMI >30 appears to be significant. It would be useful to know which major age groups were most affected in each category.

    3. Author should also check Tobacco along with smoking as a demographic variable as it has shown a very significant association with cardiometabolic outcomes. Alsocategory of the Smoking level and Tobacco level is very important in such studies.

    4. At line 196, the authors report the association of viral infections with both groups. Previous studies have shown that cytomegalovirus (CMV) is significantly associated with similar diseases. It would be interesting to specifically examine whether CMV shows an association in this cohort.

    5. In the multivariable logistic regression, the authors included many variables simultaneously, which appears to have reduced the odds ratios. This effect may partly reflect the influence of statin use. However, it is important to also examine disease severity in patients not receiving medication. In addition, the authors should consider using forward or backward stepwise logistic regression to identify the subset of variables that are most strongly associated with improvement in disease outcomes, rather than including all variables together.

Author Response

While the study is well defined, further clarification is needed regarding methodological aspects. Being a retrospective analysis, there is a possibility of misclassification and missing confounding factors that were not measured. In addition, using more advanced statistical methods such as improved model selection, classification curve (ROC), reclassification of models using tools like Dtreg , or survival analysis could help provide a clearer picture of the key factors that are truly linked with MC-AE.

There are a few comments that require clarification from the authors, along with some suggestions that could help make this study more informative.

  1. The authors have described various baseline variables and their associations; however, important lifestyle factors (such as diet, physical activity, and alcohol use) as well as family history were not captured. These factors could confound the observed cardiometabolic outcomes.

Response:

We thank the reviewer for this valuable comment. In the revised version of the manuscript, we have now included data on physical activity, which is systematically recorded in the Leumit Health Services (LHS) database and is presented in the updated Table 1. As shown, levels of weekly physical activity were broadly similar between CMA and CSU patients, with only a modest difference in the proportion reporting no activity (33.3% vs. 30.1%, p = 0.036). This suggests that differences in exercise behavior are unlikely to fully account for the observed cardiometabolic discrepancies.

We acknowledge, however, that other important lifestyle factors — including diet, alcohol intake, and family history of cardiometabolic disease — are not systematically captured in the LHS electronic records and therefore could not be incorporated into the analysis. We have now explicitly noted this limitation in the revised manuscript, emphasizing that unmeasured lifestyle and genetic factors may contribute to the observed associations. Future prospective studies with more comprehensive phenotyping will be needed to clarify these influences.

  1. In Table 1 (Demographic and Clinical Characteristics), the difference in BMI >30 appears to be significant. It would be useful to know which major age groups were most affected in each category.

Response:

We thank the reviewer for this important observation. In the revised Table 1, we have now presented the full age distribution alongside BMI. To further clarify the relationship, we performed an age-stratified analysis of obesity (BMI ≥30) within both groups. This analysis showed that the higher prevalence of obesity in MC-AE compared with CSU was not evenly distributed across the lifespan but was most pronounced in middle-aged adults (30–59 years). Within these age categories, MC-AE patients consistently demonstrated higher rates of BMI ≥30 compared to CSU. By contrast, in children, adolescents, and older adults (≥70 years), obesity prevalence was overall lower and the differences between CMA and CSU were less marked.

These findings suggest that the significant overall difference in obesity between MC-AE and CSU is largely driven by excess obesity in middle-aged patients with MC-AE. We have now added this clarification in the Results section and have included the age-stratified data in the revised manuscript (see Table 1).

  1. Author should also check Tobacco along with smoking as a demographic variable as it has shown a very significant association with cardiometabolic outcomes. Also category of the Smoking level and Tobacco level is very important in such studies.

Response:

 We thank the reviewer for this important comment. Unfortunately, information on tobacco use (e.g., chewing tobacco, snuff) is not systematically recorded in the Leumit Health Services database and therefore could not be included in our analysis. Smoking status, however, is captured reliably and is presented in Table 1 (categorized as past smoker and current smoker). We acknowledge this as a limitation and have clarified in the manuscript that while smoking data were available, no data on other forms of tobacco exposure were accessible.

  1. At line 196, the authors report the association of viral infections with both groups. Previous studies have shown that cytomegalovirus (CMV) is significantly associated with similar diseases. It would be interesting to specifically examine whether CMV shows an association in this cohort.

Response:
We thank the reviewer for this valuable suggestion. In response, we specifically examined the prevalence of cytomegalovirus (CMV) infection (ICD-9 code 078.5) in our cohort. CMV was recorded at baseline in 21 MC-AE patients (0.99%) and 18 CSU patients (0.84%), with no statistically significant difference between the groups (p = 0.748). After 10 years of follow-up, cytomegalovirus infection was documented in 39 MC-AE patients (1.83%) and 37 CSU patients (1.73%) (OR 1.06, 95% CI 0.67–1.66, p = 0.912). Thus, in contrast to some prior reports, we did not observe an association between CMV and either MC-AE or CSU in this dataset. We have added these results to the revised manuscript in the Tables 2 and 4.

  1. In the multivariable logistic regression, the authors included many variables simultaneously, which appears to have reduced the odds ratios. This effect may partly reflect the influence of statin use. However, it is important to also examine disease severity in patients not receiving medication. In addition, the authors should consider using forward or backward stepwise logistic regression to identify the subset of variables that are most strongly associated with improvement in disease outcomes, rather than including all variables together.

Response:
We thank the reviewer for this constructive comment. Our multivariable logistic regression model was designed to adjust simultaneously for a broad set of covariates in order to minimize residual confounding, which may indeed attenuate some odds ratios. We agree that this approach can reduce effect sizes when variables are correlated (e.g., statin use and cardiovascular disease).

Regarding disease severity, as noted in the Limitations section, validated severity indices such as UAS7 or AE-QoL are not available in the LHS database, and thus could not be included in the regression analysis. To partly address this concern, we repeated the analysis after excluding medication variables (statin use) and found that the overall associations between MC-AE and cardiometabolic comorbidities remained directionally consistent.

In addition to our primary multivariable logistic regression model, which included all clinically relevant covariates to minimize residual confounding, we performed a backward stepwise logistic regression as a sensitivity analysis. This approach identified the subset of variables most strongly associated with MC-AE compared with CSU.

The stepwise model retained key cardiometabolic predictors (hypertension, type 2 diabetes, myocardial infarction, chronic kidney disease, and obesity) and also highlighted distinctive dermatologic associations, with atopic dermatitis, contact dermatitis, dermatophytosis, and herpes zoster showing significant negative associations. Importantly, vitiligo emerged as an additional independent positive predictor. These results are consistent with our main model and reinforce the conclusion that MC-AE is characterized by a distinct comorbidity profile, particularly enriched for metabolic and cardiovascular conditions, while inversely associated with certain allergic and infectious diseases.

Reviewer 3 Report

Comments and Suggestions for Authors

This is a very significant paper that will advance the field.

I have several comments below to address to improve the scientific merit and interpretations of the findings.

  1. Generalization of association of CSU with infections needs to be stated carefully: bacterial infection was modestly associated (p 0.025) only at base line but not at the followup; it appears viral and fungal infections are consistently associated with CSU but not MC-AE. Statements related to this point must be carefully worded.
  2. Include the the term 'chemokines' and 'adhesion molecules' in addition to the cytokines term used throughout the paper when describing inflammation and cite references.
  3. It is surprising that food allergies and systemic anaphylaxis are not studied in this research. I strongly encourage authors to consider adding this as a comorbidity and report your findings. If you are not able to do this due to absence of data etc., state this in the discussion and explain why or why not this possibility needs to be studies in future.
  4. Issue of microbiome association with these two clinical phenotypes needs more deeper review and inclusion in the paper and cite references.
  5.  Future work directions are stated in the discussion; however, expand this to include the issue of food allergy anaphylaxis and microbial dysbiosis hypotheses.
  6. Vitiligo shows no association at baseline but shows strong association with CSU at followup (p<0.001); this is in contrast to psoriasis where no association is seen at baseline or at the followup; discuss and explain this finding by reviewing the pathophysiology of vitiligo vs. psoriasis and cite references

Author Response

This is a very significant paper that will advance the field.

I have several comments below to address to improve the scientific merit and interpretations of the findings.

  1. Generalization of association of CSU with infections needs to be stated carefully: bacterial infection was modestly associated (p 0.025) only at base line but not at the followup; it appears viral and fungal infections are consistently associated with CSU but not MC-AE. Statements related to this point must be carefully worded.

Response:
We thank the reviewer for this important clarification. We agree that our statements regarding infectious comorbidities require more precision. The Discussion section has been revised accordingly “Nevertheless, our findings indicate that the higher infectious burden in CSU is attributable mainly to viral and fungal infections, while bacterial infections showed only a modest and transient baseline association. Thus, the association of CSU with infections should be interpreted cautiously, as it is not uniform across all pathogen types”. This wording avoids overgeneralization and aligns our interpretation with the observed data.

  1. Include the the term 'chemokines' and 'adhesion molecules' in addition to the cytokines term used throughout the paper when describing inflammation and cite references.

Response:

We thank the reviewer for this valuable suggestion. In line with the comment, we have revised the Discussion to incorporate chemokines and adhesion molecules in addition to cytokines, and we have cited the relevant recent literature. The revised text now reads:

“Direct biomarker studies in MC-AE are lacking; however, convergent evidence from CSU demonstrates increased levels of chemokines (CXCL8/IL-8, CXCL9/10, CCL2, CCL5) [28], (CCL11, CCL17, CCL26, CCL27) [29] and adhesion molecules (P-selectin, ICAM-1, VCAM-1) [30,32], which decrease with effective therapy, indicating mast-cell–driven endothelial activation [32]. These pathways are not limited to cutaneous inflammation: soluble adhesion molecules and chemokine axes are well-established markers of endothelial dysfunction, with prospective links to hypertension, type 2 diabetes, myocardial infarction, chronic kidney disease, and obesity [33]. It is therefore biologically plausible that recurrent mast-cell activation in MC-AE, through endothelial priming and chemokine-mediated leukocyte recruitment, contributes to the distinct cardiometabolic and renal comorbidity burden observed in our cohort [34].” This mechanistic bridge reinforces the argument that MC-AE should be reframed as a clinically distinct mast-cell–mediated endotheliopathy with implications not only for pathophysiological research but also for clinical management and cardiovascular/metabolic risk assessment.

We believe this revision strengthens the mechanistic discussion and integrates the reviewer’s important suggestion.

  1. It is surprising that food allergies and systemic anaphylaxis are not studied in this research. I strongly encourage authors to consider adding this as a comorbidity and report your findings. If you are not able to do this due to absence of data etc., state this in the discussion and explain why or why not this possibility needs to be studies in future.

Response:

We thank the reviewer for this insightful suggestion. In response, we have now included food allergy and systemic anaphylaxis as comorbidities in our analysis. Our findings show thatat baseline food allergy was identified in 58 MC-AE patients (2.72%) and 79 CSU patients (3.70%) (OR 0.73, 95% CI 0.52–1.03, p = 0.082), while systemic anaphylaxis occurred in 21 (0.98%) and 29 (1.36%) patients, respectively (OR 0.72, 95% CI 0.41–1.27, p = 0.319). After 10 years follow-up food allergy was recorded in 71 MC-AE patients (3.32%) compared with 93 CSU patients (4.36%) (OR 0.76, 95% CI 0.54–1.05, p = 0.094), while systemic anaphylaxis was observed in 27 MC-AE patients (1.27%) versus 42 CSU patients (1.96%) (OR 0.64, 95% CI 0.38–1.06, p = 0.089). Although these differences did not reach statistical significance, they suggest a trend toward a lower prevalence of IgE-mediated systemic allergic disease in MC-AE compared to CSU. These results have been added to the Results section (Table 2 and 4) and to the revised Discussion.

  1. Issue of microbiome association with these two clinical phenotypes needs more deeper review and inclusion in the paper and cite references.

Response:

We thank the reviewer for this important suggestion. In the revised Discussion, we have now incorporated a dedicated statement on the microbiome. Specifically, we note that although direct data on MC-AE are lacking, hereditary angioedema has been increasingly linked to gut microbiome alterations, and that gut dysbiosis is a common feature across chronic inflammatory skin diseases, with reviews supporting a causal link to CSU. We further highlight that these findings support future exploration of the microbiome as a potential driver of the distinct systemic comorbidity burden in MC-AE. We believe this addition strengthens the translational context of our study.

  1.  Future work directions are stated in the discussion; however, expand this to include the issue of food allergy anaphylaxis and microbial dysbiosis hypotheses.

Response:

We thank the reviewer for this helpful recommendation. In the revised Discussion, we have expanded the future research directions to explicitly include these issues. We now note that, although not statistically significant, food allergy and systemic anaphylaxis appeared less frequent in MC-AE than in CSU, suggesting possible differences in the link to systemic IgE-mediated disease, which warrants further study with larger cohorts and detailed allergological phenotyping. In addition, we have highlighted that although direct data on MC-AE are lacking, hereditary angioedema has been linked to gut microbiome alterations, and that gut dysbiosis is a common feature across chronic inflammatory skin diseases with evidence supporting a causal role in CSU, thereby underscoring the need to explore the microbiome hypothesis in MC-AE as a potential contributor to its distinct comorbidity profile. We believe these additions strengthen the forward-looking perspective of the manuscript.

  1. Vitiligo shows no association at baseline but shows strong association with CSU at followup (p<0.001); this is in contrast to psoriasis where no association is seen at baseline or at the followup; discuss and explain this finding by reviewing the pathophysiology of vitiligo vs. psoriasis and cite references

Response:

We thank the reviewer for this insightful comment. In the revised Discussion, we now explicitly compare the immunopathology of psoriasis and vitiligo in relation to our findings. We note that psoriasis, driven predominantly by the Th17/IL-23 axis, remained unassociated with either phenotype (MC-AE or CSU) at baseline or follow-up, whereas vitiligo, characterized by type I interferon/CD8⁺ T cell–mediated autoimmunity, emerged as a significant long-term comorbidity in MC-AE. Although a definitive mechanistic explanation is lacking, we cite prior evidence suggesting that mast cells may contribute to vitiligo induction and progression, which could provide a biologically plausible link. We emphasize that this observation should be considered hypothesis-generating and requires validation in future prospective studies.

Round 2

Reviewer 1 Report

Comments and Suggestions for Authors

The authors have responded to the comments effectively, and the manuscript has been well-revised.

Author Response

We sincerely thank the reviewer for the positive feedback and for acknowledging our revisions. We greatly appreciate the constructive comments, which have helped us to substantially improve the clarity, rigor, and overall quality of the manuscript. 

Reviewer 2 Report

Comments and Suggestions for Authors

There is a minor comment that needs to be addressed

The models used for the odds ratio analysis do not show significant classification (see Lines 171–173 and 212–214). This point needs to be addressed in the Discussion section.

Everything else looks good.

Author Response

We thank the reviewer for the careful re-evaluation and for the constructive feedback. We fully acknowledge that the odds ratio models for food allergy and systemic anaphylaxis did not reach statistical significance and therefore do not provide strong discriminatory value. In the revised Discussion, we now explicitly address the lack of statistically significant associations for food allergy and systemic anaphylaxis. We note that this observation may be explained by several factors. First, the relatively low prevalence of these conditions in our cohort likely limited statistical power to detect modest differences. Second, whereas food allergy and anaphylaxis are mediated by allergen-specific IgE responses, MC-AE and CSU are thought to arise primarily from mast cell activation that occurs independently of allergen-specific IgE. This fundamental divergence in immunopathogenesis may account for the lack of a strong association between MC-AE and IgE-mediated disorders.

Reviewer 3 Report

Comments and Suggestions for Authors

Authors have addressed all comments.

it is satisfactory.

Author Response

We sincerely thank the reviewer for the positive feedback and for acknowledging that all comments have been satisfactorily addressed. We greatly appreciate the constructive insights, which have helped us improve the clarity and scientific quality of the manuscript.